# Evaluating the Effectiveness of Nirsevimab in Reducing Pediatric RSV Hospitalizations in Spain

**DOI:** 10.3390/vaccines12101160

**Published:** 2024-10-11

**Authors:** Sara Jimeno Ruiz, Adrián Peláez, Agustín Labourt, Florencia Mercedes Acuña, Lucía Linares, Isabel Llana Martín, Ángeles Calle Gómez, Silvina Natalini Martínez

**Affiliations:** 1Departamento de Pediatría, Hospital HM Puerta del Sur, HM Hospitales, 28938 Madrid, Spain; sarajimeno@hotmail.com (S.J.R.); agustinlabourt@gmail.com (A.L.); florenciamacuna@gmail.com (F.M.A.); lulalinares@gmail.com (L.L.); illana@hmhospitales.com (I.L.M.); angelescalle@hotmail.com (Á.C.G.); slnatalini@hmhospitales.com (S.N.M.); 2Unidad de Vacunas, HM Hospitales, 28938 Madrid, Spain; 3Facultad de Ciencias de la Salud-HM Hospitales, Universidad Camilo José Cela, 28692 Madrid, Spain; 4Instituto de Investigación Sanitaria HM Hospitales, 28015 Madrid, Spain; 5Departamento de Unidad de Análisis de Datos, Fundación de Investigación HM Hospitales, Facultad de Ciencias de la Salud-HM Hospitales, Universidad Camilo José Cela, 28692 Madrid, Spain; 6Centro de Investigación en Red de Enfermedades Respiratorias (CIBERES), Instituto de Salud Carlos III (ISCIII), 28029 Madrid, Spain

**Keywords:** respiratory syncytial virus (RSV), Nirsevimab, pediatric respiratory care, RSV-related lower respiratory tract infection, healthcare burden reduction

## Abstract

**Background/Objectives**: Respiratory syncytial virus (RSV) is a major cause of hospitalization in infants. Nirsevimab has demonstrated to be a promising tool for preventing severe RSV disease. Although clinical trials have demonstrated the efficacy of Nirsevimab in preventing severe RSV disease, evidence regarding its performance in real-world clinical settings is still limited due to its recent introduction. This study aims to fill this knowledge gap by evaluating the impact of Nirsevimab in a cohort of infants and determining its effectiveness in reducing the burden of RSV disease. **Methods**: A retrospective study of RSV hospitalizations was conducted in children under six months of age, between 1 October and 31 March, across four seasons: pre-COVID (2018–2019), COVID (2019–2020), post-COVID pre-Nirsevimab (2022–2023), and Nirsevimab season (2023–2024). **Results**: Nirsevimab demonstrated significant efficacy in reducing RSV-related hospitalizations in infants under six months of age. During the 2023/2024 season, following the introduction of Nirsevimab, there was a substantial reduction in RSV-related lower respiratory tract infection (LRTI) hospitalizations. Among infants under 3 months of age, hospitalizations decreased by 79.3% (IRR: 0.21, 95% CI: 0.12–0.34). In infants aged 3 to 6 months, there was a 66.9% reduction (IRR: 0.33, 95% CI: 0.15–0.64). Additionally, Nirsevimab decreased the severity of RSV cases with LRTI who required the support of equipment for sanitary use, further reducing overall healthcare burden. **Conclusions**: These results underscore Nirsevimab’s vital role in preventing severe RSV infections and hospitalizations, especially among the most vulnerable infants, positioning it as a critical advancement in pediatric respiratory care.

## 1. Introduction

Respiratory syncytial virus (RSV) is a common respiratory virus with a global distribution and a marked seasonal pattern [1]. For decades, it has been the leading cause of respiratory infections worldwide during epidemic periods, with peak incidence in the Northern Hemisphere occurring between November and February [1].

RSV poses a substantial burden on healthcare systems. Estimates suggest that, globally, approximately 34 million new cases of lower respiratory tract infections (LTRI) in children under five years are attributed to RSV, leading to 3.4 million hospitalizations and around 199,000 deaths annually, predominantly in developing countries [2,3]. In Spain, RSV-related hospitalizations represent an estimated annual cost of approximately 50 million euros to the National Health System [4]. The morbidity associated with RSV outside the hospital setting is less quantifiable, with estimates suggesting that more than a quarter of the total direct healthcare expenditure could be independent of hospitalization [5,6,7].

RSV infection affects 60–70% of children within their first year of life and more than 80% of those under two years old, also serving as a significant cause of respiratory exacerbations in older children and adults [8]. In recent years, RSV infections have become a major reason for seeking care in primary healthcare and pediatric emergency settings. During epidemic periods, RSV has emerged as the primary cause of hospitalization, accounting for 70–85% of admissions for bronchiolitis [2,8].

After many years of research, we now have preventive measures against RSV infections that have significantly impacted disease incidence in both children and adults. Nirsevimab (Beyfortus, AstraZeneca, and Sanofi), a long-acting monoclonal antibody against RSV, was approved in Europe (31 October 2022), the United Kingdom (9 November 2022), Canada (19 April 2023), and the United States (17 July 2023) [9,10]. The MELODY trial reported an efficacy of Nirsevimab of 76.8% against hospitalization for RSV-related lower respiratory tract infections (LRTI) and 76.4% against non-hospitalized LRTI cases [9,11,12]. The HARMONIE trial demonstrated an efficacy of 83.0% against RSV-related LRTI hospitalizations and 76.0% against severe LRTI in preterm infants (<29 weeks) during their first RSV season [9,13]. Currently, Nirsevimab has become the first immunization product for universal prophylaxis against RSV in infants [9,10]. 

While clinical trials have demonstrated the efficacy of Nirsevimab, real-world evidence is still limited. For this reason, some very recent studies are providing preliminary results on the effectiveness and impact of the use of Nirsevimab in the 2023–2024 season in several regions of Spain [9,14,15,16] and worldwide [17,18]. 

This study aims to evaluate the real-world effectiveness of Nirsevimab in preventing hospitalizations due to RSV infections among infants in different epidemic seasons.

## 2. Materials and Methods

A retrospective multicenter observational study of pediatric patients diagnosed with RSV infection during four distinct epidemic periods time periods was conducted. These periods span between 1 October 2018 and 31 March 2019 (designated as pre-COVID season), between 1 October 2019 and 31 March 2020 (COVID season), between 1 October and 31 March 2022–2023 (Post-COVID season prior to Nirsevimab), and between 1 October and 31 March 2023–2024 (Nirsevimab season).

Patients less than 6 months old at the beginning of the epidemic period were selected as inclusion criteria (1 October of each season respectively), as well as those born during the epidemic period (1 October to 31 March of each season) who were hospitalized for RSV infection confirmed by antigenic testing and/or polymerase chain reaction (PCR). Patients who did not meet these inclusion criteria were excluded.

Several sociodemographic and clinical variables were collected, as well as the results of antigenic and polymerase chain reaction (PCR) tests for respiratory viruses. In addition, the different pharmacological treatments used, oxygen therapy requirements, and the need for invasive or non-invasive ventilation were collected. The variables were extracted from the Doctoris computerized medical record, with all patient identifiers previously anonymized to protect confidentiality and privacy. The study protocol was approved by the local ethics committee approval number 24.06.2359. The document followed STROBE (http://www.strobe-statement.org/) recommendations for observational studies.

To determine the minimum sample size required to assess the vaccine’s effect (with a 75% risk reduction), a calculation was performed with a significance level of 1% and a statistical power of 99%. The calculation indicated that 68 patients were needed. However, to account for potential participant dropouts and ensure the robustness of the model, the sample was increased to include the maximum number of patients available in our database, resulting in a total of 646 patients.

A person-time approach was used for each individual during statistical analyses. A preliminary descriptive analysis of patient characteristics was performed, calculating measures of central tendency and dispersion for quantitative variables, as well as counts and percentages for qualitative variables. Differences between groups were assessed using normality tests (Shapiro–Wilk, Kolmogorov–Smirnov) and homogeneity tests (Levene), with parametric (ANOVA, Student’s *t*-test) or non-parametric (Kruskal–Wallis, Wilcoxon rank-sum) tests applied as appropriate. To reduce the risk of Type I errors due to multiple comparisons, the Bonferroni correction was applied where necessary. For qualitative variables, χ^2^ or Fisher’s exact tests were used. 

Poisson regression models with robust variance and Cox proportional hazards models, adjusted, were employed. The association of various sociodemographic and comorbidities covariates with the outcome variable was evaluated in univariate analyses, and those with significant results were included in the multivariable model. The efficacy of Nirsevimab against RSV hospitalization was calculated as (1 − point estimate) × 100, with 95% confidence intervals (CIs). Only participants with complete data and follow-up times greater than zero were included. Hospitalization rates, both overall and by age group (<3 months and 3–6 months) per 10,000 hospitalized individuals in the same age group, were calculated for each of the four study seasons. The incidence ratio was obtained by comparing these rates to the Post-COVID prior to Nirsevimab season, and the percentage change was calculated as (rate difference/Post-COVID prior to Nirsevimab rate) × 100. Data analysis and visualization were conducted using RStudio version 4.3.1.

## 3. Results

### 3.1. Sociodemographic, Clinical, and Treatment Data across Different RSV Seasons

A total of 646 unique patients were included in the study, distributed as follows: 292 (45.2%) in the pre-COVID season, 78 (12.1%) in the COVID season, 199 (30.8%) in the post-COVID season prior to Nirsevimab, and 77 (11.9%) in the Nirsevimab season. Regarding Nirsevimab administration in the 77 patients analyzed during the Nirsevimab season (Figure A1), 26 (33.8%) received Nirsevimab, and 51 (66.2%) did not. 

These 646 unique patients were drawn from HM Modelo-HM Belén hospitals (73 patients), HM Montepríncipe (215 patients), HM Nens (81 patients), and HM Puerta del Sur (277 patients). A descriptive analysis was performed on the sociodemographic, clinical, and treatment data of the patients, comparing the four RSV seasons (Table 1). The overall mean age was 4.2 months (±3.6). Patients in the Nirsevimab season had a significantly higher mean age (6.4 months ± 4.3) (*p* < 0.001). A notable reduction in the number of patients under 3 months and those aged 3–6 months was observed during the Nirsevimab season. Regarding the length of hospital and ICU stays, significant differences were found, with shorter stays observed in the Nirsevimab and pre-COVID seasons, respectively (*p* ≤ 0.002). Concerning the distribution of comorbidities, only respiratory and bronchial pathologies, as well as low birth weight, showed significant differences during the Nirsevimab season (*p* ≤ 0.048).

### 3.2. Hospitalization Patterns in Children under 1 Year Due to RSV Infections across Different Seasons

In Figure 1, four distinct patterns in the hospitalization rate due to RSV infections in children under 1 year of age (per 10,000 children) can be observed. Firstly, from the 2018/2019 season (October–March) to the 2020/2021 season, a clear seasonal trend in hospitalization rates is evident. This behavior corresponds to the pre-COVID period and is consistent across all age groups analyzed (Figure 1B,C). Between the 2020/2021 and 2021/2022 seasons, the pattern becomes chaotic, with no defined seasonality. This change aligns with the COVID period, and the group of hospitalized children over 6 months shows fewer peak values compared to other age groups. In the 2022/2023 season, marking the post-COVID period and prior to the Nirsevimab period, seasonality is restored both in the overall patient group and across the three age groups, resembling the pattern observed in the initial season. A notable peak in hospitalization is seen in patients under 3 months of age. Finally, in the 2023/2024 season, corresponding to the Nirsevimab period, another seasonal RSV peak is observed across all groups. However, this peak is lower for children under 3 months and those aged 3 to 6 months compared to previous seasonal peaks. Nevertheless, in patients over 6 months, the peak value remains similar to that observed in earlier seasons.

### 3.3. Impact of Nirsevimab on Hospitalization Rates for RSV across Different Seasons

Hospitalization rates for LRTI related to RSV were analyzed across different pandemic periods, along with the impact of Nirsevimab on these events (Table 2; Figure 2). Hospitalization rates were expressed per 10,000 population and broken down by age group, covering all ages and two specific categories: under 3 months and from 3 to 6 months. In the pre-COVID season, the overall hospitalization rate for RSV was 9129 [95% CI: 8699–9453] per 10,000 population. Infants aged between 3 to 6 months group had the highest rate at 9263 [95% CI: 8541–9699], while the rate for infants under 3 months had the highest rate at 9041 [95% CI: 8443–9466]. During the COVID season, the overall hospitalization rate decreased to 6154 [95% CI: 5075–7155] per 10,000 population. Infants under 3 months showed a reduction, with a rate of 6327 [95% CI: 4829–7658], and the 3 to 6 months group had a rate of 5952 [95% CI: 4328–7437]. In the post-COVID season prior to Nirsevimab, the overall hospitalization rate increased to 7540 [95% CI: 6859–8139] per 10,000 population. Infants under 3 months had a rate of 7838 [95% CI: 6956–8563], and the 3 to 6 months group had a rate of 7105 [95% CI: 5951–8089]. With the introduction of Nirsevimab, the overall hospitalization rate decreased significantly to 3425 [95% CI: 2353–4628] per 10,000 population. Infants under 3 months had a rate of 3636 [95% CI: 2241–5223], while the 3 to 6 months group had a rate of 3103 [95% CI: 1528–5083]. The relative reduction in hospitalization rates compared to the pre-COVID season showed a significant decrease with the administration of Nirsevimab. For all ages, the reduction was 54.6%. When evaluated by age groups, the 3 to 6 months group showed the largest reduction (56.3%), followed by the group of infants under 3 months (53.6%).

### 3.4. Efficacy of Nirsevimab against RSV Severity

The efficacy of Nirsevimab against various severity criteria for LRTI related to RSV was evaluated over two seasons: 2022–2023 (without receiving Nirsevimab) and 2023–2024 (receiving Nirsevimab) (Table 3; Figure 3). Efficacy was assessed using Poisson regression models, which allow for the calculation of adjusted IRR and estimate efficacy in percentage terms. The adjustment was performed for the set of variables, including sociodemographic and comorbidities that were associated in each case with the response variable in a univariate approach. In infants under 3 months, there was a significant 79.3% reduction in RSV-related LRTI hospitalizations (IRR: 0.21, 95% CI: 0.12–0.34), while in those aged 3–6 months, the reduction was 66.9% (IRR: 0.33, 95% CI: 0.15–0.64). For RSV-related LRTI hospitalizations requiring oxygen support, the reduction was 77.3% for infants under 3 months (IRR: 0.23, 95% CI: 0.11–0.42) and 91.7% for those aged 3–6 months (IRR: 0.08, 95% CI: 0.01–0.39). Hospitalizations involving ICU admission showed a 68.7% reduction in those under 3 months (IRR: 0.32, 95% CI: 0.16–0.56), and a 67.9% reduction in the 3–6-month group (IRR: 0.32, 95% CI: 0.04–1.20). For LRTI hospitalizations requiring NIMV and/or HFNO, a 78.0% reduction was observed in the under 3 months group (IRR: 0.22, 95% CI: 0.11–0.41) and a 91.4% reduction in the 3–6-month group (IRR: 0.09, 95% CI: 0.01–0.41). For cases requiring IMV, the reduction was 87.2% in infants under 3 months (IRR: 0.13, 95% CI: 0.01–0.68), while no events were recorded in the 3–6-month group. Finally, RSV-related LRTI hospitalizations with all possible complications showed a 77.3% reduction in infants under 3 months (IRR: 0.23, 95% CI: 0.12–0.40) and an 83.4% reduction in those aged 3–6 months (IRR: 0.17, 95% CI: 0.03–0.56). Finally, for RSV hospitalizations with all possible complications, a reduction of 75.8% was observed in infants under 3 months (IRR: 0.24, 95% CI: 0.14–0.39), while the reduction was 64.5% in the 3–6-month group (IRR: 0.36, 95% CI: 0.17–0.67).

## 4. Discussion

RSV remains a leading cause of acute respiratory infections in infants, with high rates of hospitalization, especially during the first six months of life [8,19,20]. This virus places a considerable burden on healthcare systems, with significant challenges in terms of costs and resources [2,6]. Although there have been continuous efforts to mitigate its impact, effective prevention options remains limited. The recent introduction of Nirsevimab, a long-acting monoclonal antibody, offers new hope to reduce this burden of hospitalizations [11,12,13,20]. The results of the MELODY, HARMONIE, and NIRSE-GAL studies support its efficacy [9,10,11,12,13]. Spain has been one of the first countries to recommend Nirsevimab prophylaxis for the 2023–2024 season [21]. More than 200,000 doses of Nirsevimab have been administered during the 2023–2024 season, achieving an average coverage rate of 91.9% in newborns (range 85.7–96.7%) [22]. Notably, none of the premature infants received palivizumab during the 2023–2024 season, in accordance with the recommendations for the prophylaxis of severe RSV infections. However, due to its recent novelty, its efficacy in actual clinical practice has yet to be confirmed, as clinical trials are conducted under controlled conditions that may not fully reflect the complexities of everyday clinical settings. This study aims to provide a more accurate picture of the efficacy of Nirsevimab in preventing RSV-related hospitalizations in infants during epidemic seasons, both before and after its recommendation for use.

Our results show that the introduction of Nirsevimab in the 2023/2024 season has achieved a significant reduction in RSV LRTI-related hospitalizations in infants younger than one year, with a particularly notable impact on the pediatric population younger than three months. This is of great importance due to the increased bronchial hyperreactivity in this age group and the serious long-term complications it can generate [23,24]. This decrease is remarkable given the history of high hospitalization rates in previous seasons (Figure 1). However, this decrease is only observed in the age group 3 months and 3–6 months. This effect may be related to the fact that infants older than 6 months received the vaccine when the season had already begun, increasing the risk of having been infected by RSV before receiving the antibodies and, consequently, a greater number of hospitalizations. These results underscore the efficacy of Nirsevimab in the first months of life, a critical period when infants are most vulnerable to serious RSV complications.

Comparing hospitalization rates during the post-COVID Nirsevimab season with the post-COVID period prior to the Nirsevimab season, a consistent reduction is observed, with greater protection in infants younger than 6 months: a decrease of 56.3% in the 3–6-month group and 53.6% in those younger than 3 months (Table 2; Figure 2). The implementation of Nirsevimab has proven highly effective not only in reducing the rate of hospitalizations due to severe respiratory infections caused by RSV but also in decreasing the severity of these hospitalizations (Table 3; Figure 3). This suggests that the benefit of Nirsevimab extends beyond respiratory infections, contributing to an overall reduction in the hospitalization burden.

This study presents several key strengths that reinforce the robustness of its findings. The multicenter design, which includes multiple hospitals and regions, ensures greater representativeness and generalizability of the results at the population level. Furthermore, the comparison between different periods, encompassing both pre- and post-COVID seasons, allows for the contextualization of Nirsevimab’s effects in various epidemiological scenarios, especially in the context of the COVID-19 pandemic. We emphasize how viral interactions, competition for ecological niches, and non-pharmaceutical interventions impacted viral epidemiology during the pandemic.

The pandemic has altered the epidemiology of respiratory infections, and understanding these changes is essential for effective public health planning [25,26]. The detailed assessment of different levels of severity of hospitalization, from different modalities of oxygen therapy and respiratory support to intensive care, provides a complete picture of the clinical impact of Nirsevimab. However, this study also faces some limitations that should be considered when interpreting its results. As a retrospective study, it is subject to potential selection biases and errors in data collection, which could compromise the accuracy of the conclusions. Although several variables were controlled in the analyses, unmeasured confounding factors that could have influenced the results cannot be ruled out. Additionally, although clinical trials have demonstrated the efficacy of Nirsevimab in preventing severe RSV disease, evidence of its performance in real clinical settings remains limited due to its recent introduction. In addition, we also recognize the potential limitation of varying RSV strains across seasons and how this could influence the results.

Our study shows a similar efficacy to that described in the literature [9,17], which supports our continued focus on achieving adequate protection for the population most vulnerable to RSV. However, it is essential to conduct additional long-term studies in different clinical settings and different population groups to validate the efficacy of Nirsevimab and better understand its impact on RSV prevention in real-world settings. This fact will help to implement appropriate interventions, such as the use of Nirsevimab in vulnerable population groups or RSV vaccination in pregnant women, to mitigate the burden on healthcare systems.

## 5. Conclusions

In conclusion, the recent introduction of Nirsevimab has demonstrated a promising reduction in RSV-related hospitalizations, particularly in infants younger than six months, as well as a decrease in the severity of hospitalizations. Continued monitoring and research are essential to fully understand its long-term impact and to refine preventive strategies in the pediatric population.

## Figures and Tables

**Figure 1 vaccines-12-01160-f001:**
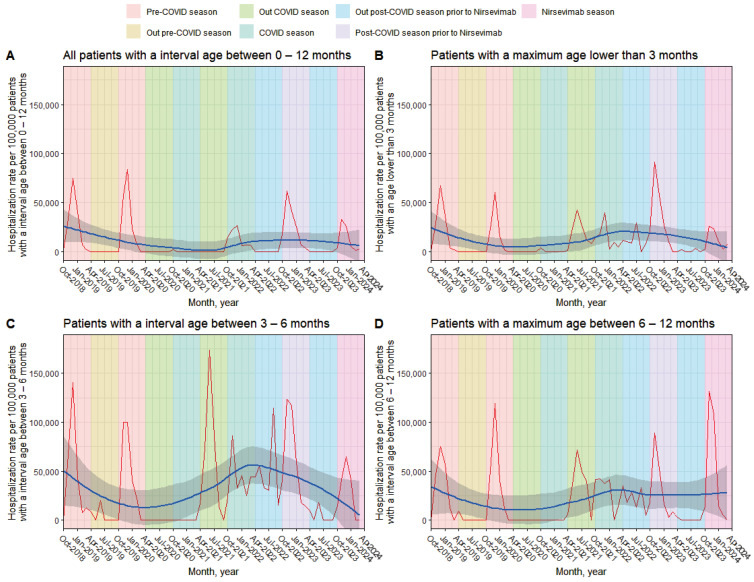
RSV hospitalization rates in children under 1 year of age (per 10,000 children). The shaded areas in different colors correspond to the seasons (1 October to 31 March) and between seasons (1 April to 30 September) of RSV before, during, and post-pandemic COVID-19 (2020–2022). In addition, the post-Nirsevimab season (2023–2024) was included. Infection rates were plotted for all children younger than 12 months (**A**), younger than 3 months (**B**), between 3 and 6 months (**C**), and older than 6 months (**D**). A smoothing line was added to the graph to help visualize trends in the data.

**Figure 2 vaccines-12-01160-f002:**
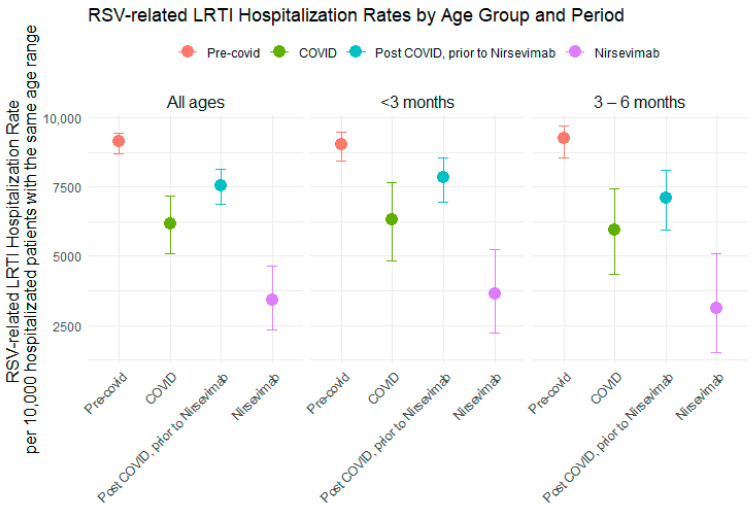
RSV-related hospitalization rates by age group and period. Each panel represents a specific age group. Dots indicate the hospitalization rates per 10,000 hospitalized patients with the same age range, with error bars showing the 95% CI.

**Figure 3 vaccines-12-01160-f003:**
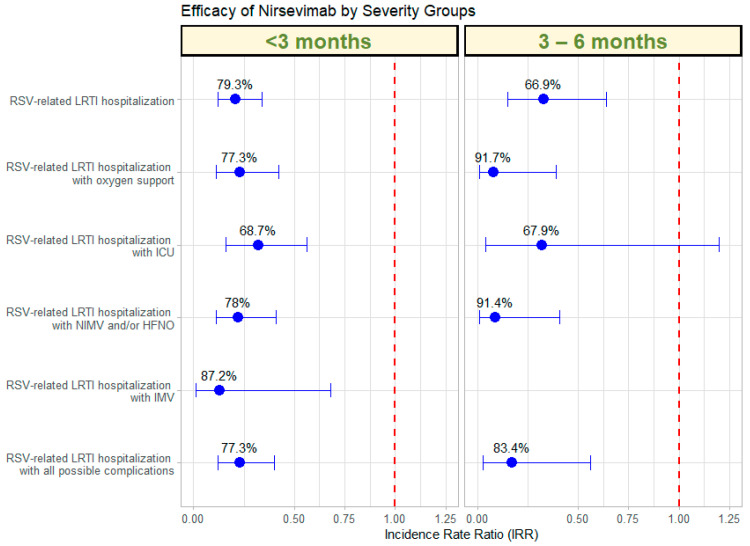
Efficacy of Nirsevimab by severity group, represented by the IRR. Each point shows the IRR for a specific severity item, with error bars indicating 95% CI. The effectiveness percentages are labelled next to each point.

**Table 1 vaccines-12-01160-t001:** Overall descriptive and comparative sociodemographic, clinical, and treatment data in total and for each of the RSV seasons.

Item	Total (*n* = 646)	Pre-COVID (*n*1 = 292)	COVID (*n*2 = 78)	Post-COVID Prior to Nirsevimab (*n*3 = 199)	Nirsevimab (*n*4 = 77)	*p*-Value
Age (months)	4.2 (±3.6)	3.8 (±3.3)	4.1 (±3.3)	4.0 (±3.5)	6.4 (±4.3) ↑	<0.001
≤3 months	281 (43.5%)	136 (46.6%)	34 (43.6%)	89 (44.7%)	22 (28.6%) ↓	<0.001
3–6 months	188 (29.1%)	92 (31.5%)	26 (33.3%)	59 (29.6%)	11 (14.3%) ↓	0.021
>6 months	177 (27.4%)	64 (21.9%)	18 (23.1%)	51 (25.6%)	44 (57.1%) ↑	0.024
Sex [Male]	356 (55.1%)	155 (53.1%)	48 (61.5%)	117 (58.8%)	36 (46.8%)	0.168
Stay	5.6 (±4.6)	5.6 (±3.6)	5.2 (±3.9)	6.1 (±5.0)	5.1 (±6.8) ↓	0.011
<10 days	550 (85.1%)	249 (85.3%)	66 (84.6%)	166 (83.4%)	69 (89.6%) ↑	<0.001
10–30 days	94 (14.6%)	43 (14.7%)	12 (15.4%)	32 (16.1%)	7 (9.1%) ↓	0.009
>30 days	2 (0.3%)	0 (0.0%)	0 (0.0%)	1 (0.5%)	1 (1.3%)	0.538
ICU	197 (30.5%)	95 (32.5%)	20 (25.6%)	62 (31.2%)	20 (26.0%)	0.527
ICU Stay	5.3 (4.3)	4.1 (±3.0) ↓	6.0 (±3.8)	6.5 (±5.0)	6.6 (±6.4)	0.004
<10 days	169 (86.2%)	91 (95.8%) ↑	16 (80.0%)	47 (77.0%)	15 (75.0%)	0.002
10–30 days	27 (13.8%)	4 (4.2%) ↓	4 (20.0%)	14 (23.0%)	5 (25.0%)	0.003
>30 days	0 (0%)	0 (0%)	0 (0%)	0 (0%)	0 (0%)	1.000
Comorbidities						
Neonatal respiratory problems	96 (14.9%)	45 (15.4%)	7 (9.0%)	31 (15.6%)	13 (16.9%)	0.469
APLV	4 (0.6%)	2 (0.7%)	0 (0%)	1 (0.5%)	1 (1.3%)	0.770
Atopy	3 (0.5%)	2 (0.7%)	0 (0%)	0 (0%)	1 (1.3%)	0.430
Cardiopathies	9 (1.4%)	2 (0.7%)	0 (0%)	4 (2.0%)	3 (3.9%)	0.101
Syndromes	2 (0.3%)	0 (0.0%)	0 (0.0%)	1 (0.5%)	1 (1.3%)	0.280
Immunodeficiencies	3 (0.5%)	1 (0.3%)	0 (0%)	1 (0.5%)	1 (1.3%)	0.654
Twins	7 (1.1%)	6 (2.1%)	0 (0%)	0 (0%)	1 (1.3%)	0.131
Premature	30 (4.6%)	16 (5.5%)	1 (1.3%)	7 (3.5%)	6 (7.8%)	0.191
Low birth weight	19 (2.9%)	6 (2.1%)	1 (1.3%)	6 (3.0%)	6 (7.8%) ↑	0.048

Data are shown as *n* (%), mean ± sd. “The direction of the significant post-hoc pairwise comparison is indicated by an arrow (↑, ↓)”.

**Table 2 vaccines-12-01160-t002:** Impact of Nirsevimab on RSV-related LRTI hospitalizations at each stage overall and separated by age group.

RSV-Related LRTI Hospitalization Rate Per 10,000	Pre-COVID	COVID	Post-COVID Prior to Nirsevimab	Nirsevimab
All ages	9129 [8699–9453]	6154 [5075–7155]	7540 [6859–8139]	3425 [2353–4628]
<3 months	9041 [8443–9466]	6327 [4829–7658]	7838 [6956–8563]	3636 [2241–5223]
3–6 months	9263 [8541–9699]	5952 [4328–7437]	7105 [5951–8089]	3103 [1528–5083]
Ratio against reference season				
All ages	1.2 [1.1–1.4]	0.8 [0.6–1.0]	Ref	0.5 [0.3–0.7]
<3 months	1.2 [1.0–1.4]	0.8 [0.6–1.1]	Ref	0.5 [0.3–0.8]
3–6 months	1.3 [1.1–1.6]	0.8 [0.5–1.2]	Ref	0.4 [0.2–0.9]
Relative reduction (%)				
All ages	+21.1%	−18.4%	Ref	−54.6%
<3 months	+15.4%	−19.3%	Ref	−53.6%
3–6 months	+30.4%	−16.2%	Ref	−56.3%

Data are shown as rate and 95% CI. Ref: reference group, when comparing.

**Table 3 vaccines-12-01160-t003:** Efficacy of Nirsevimab against different severity criteria using Poisson regression models.

Severity Groups	Incidence Rate Ratio Adjusted (95% CI)	Effectiveness Adjusted% (95% CI)
RSV-related LRTI hospitalization		
<3 months	0.21 (0.12–0.34)	79.3% (65.7–88.3%)
3–6 months	0.33 (0.15– 0.64)	66.9% (36.2–84.7%)
RSV-related LRTI hospitalization with oxygen support		
<3 months	0.23 (0.11–0.42)	77.3% (58.3–88.8%)
3–6 months	0.08 (0.01–0.39)	91.7% (60.8–99.5%)
RSV-related LRTI hospitalization with ICU		
<3 months	0.32 (0.16–0.56)	68.7% (43.7–83.8%)
3–6 months	0.32 (0.04–1.20)	67.9% (−20.2–95.0%)
RSV-related LRTI hospitalization with NIMV and/or HFNO		
<3 months	0.22 (0.11–0.41)	78.0% (59.5–89.1%)
3–6 months	0.09 (0.01–0.41)	91.4 (59.0–99.5%)
RSV-related LRTI hospitalization with IMV		
<3 months	0.13 (0.01–0.68)	87.2% (31.8–99.3%)
3–6 months	NA	NA
RSV-related LRTI hospitalization with all possible complications		
<3 months	0.23 (0.12–0.40)	77.3% (60.1–88.1%)
3–6 months	0.17 (0.03–0.56)	83.4% (44.2–97.3)

The efficacy of Nirsevimab was estimated from incidence rate ratios calculated using Poisson regression models, which were adjusted for season group, sex, age, hospital, previous infections, immunodeficiencies, heart disease, prematurity, neonatal respiratory problems, syndromes, and twins. Only patients with non-zero follow-up times were included. All possible complications: Oxygen support, ICU, NIMV, and IMV. NA: not available.

## Data Availability

The data supporting the reported results cannot be shared due to ethical restrictions.

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
