# Peer review of "Evaluating the Effectiveness of Nirsevimab in Reducing Pediatric RSV Hospitalizations in Spain"

_vaccines, 2024, doi:10.3390/vaccines12101160_

Round 1

Reviewer 1 Report

Comments and Suggestions for Authors

This manuscript presents a retrospective study investigating the real-world effectiveness of Nirsevimab, a monoclonal antibody against respiratory syncytial virus (RSV), in reducing infant hospitalizations. Given the recent introduction of Nirsevimab and the limited real-world data available, this research addresses a gap in knowledge regarding pediatric respiratory care. The authors analyze data spanning four RSV seasons, including pre-COVID, COVID, post-COVID pre-Nirsevimab, and the Nirsevimab season (2023-2024), providing insights into the antibody's impact in diverse epidemiological contexts. The study focuses on a cohort of infants under six months of age, a particularly vulnerable population for severe RSV disease. Preliminary findings suggest a significant reduction in RSV-related hospitalizations following the introduction of Nirsevimab, particularly among infants under three months of age. This research has the potential to inform clinical practice and public health policy regarding RSV prevention strategies.

The authors used a combination of appropriate statistical techniques to analyze their data and assess the efficacy of Nirsevimab in preventing RSV hospitalization. The use of robust methods and adjustment for potential confounding factors strengthens the validity of their findings.

The manuscript is well constructed, and I believe the study is sound. I recommend publication after minor revisions. Specifically:

Figure 1 appears to be missing from this manuscript. 

In the study design in the manuscript, "The efficacy of Nirsevimab against various severity criteria for LRTI related to RSV was evaluated over two seasons: 2022-2023 (without receiving Nirsevimab) and 2023-2024 (receiving Nirsevimab) (Table 3; Figure 3)." While both periods are classified as post-COVID, the circulating variants during each season may be different, which could act as a confounding variable in assessing the severity of RSV and, consequently, the efficacy of the drug. This variation could potentially affect the results of the drug's efficacy. It may be worth commenting on this potential limitation in the manuscript. 

Author Response

We sincerely appreciate your thoughtful review of our manuscript. Your recognition of our study's contribution to understanding the real-world effectiveness of Nirsevimab in reducing infant hospitalizations is invaluable. We are grateful for your acknowledgment of the significance of our research, particularly regarding its potential to inform clinical practice and public health policy in the context of RSV prevention strategies. Your comments regarding our methodological approach and statistical techniques further validate our efforts and motivate us to enhance our work. Thank you for your support and constructive feedback.

Reviewer 1

This manuscript presents a retrospective study investigating the real-world effectiveness of Nirsevimab, a monoclonal antibody against respiratory syncytial virus (RSV), in reducing infant hospitalizations. Given the recent introduction of Nirsevimab and the limited real-world data available, this research addresses a gap in knowledge regarding pediatric respiratory care. The authors analyze data spanning four RSV seasons, including pre-COVID, COVID, post-COVID pre-Nirsevimab, and the Nirsevimab season (2023-2024), providing insights into the antibody's impact in diverse epidemiological contexts. The study focuses on a cohort of infants under six months of age, a particularly vulnerable population for severe RSV disease. Preliminary findings suggest a significant reduction in RSV-related hospitalizations following the introduction of Nirsevimab, particularly among infants under three months of age. This research has the potential to inform clinical practice and public health policy regarding RSV prevention strategies.

The authors used a combination of appropriate statistical techniques to analyze their data and assess the efficacy of Nirsevimab in preventing RSV hospitalization. The use of robust methods and adjustment for potential confounding factors strengthens the validity of their findings.

The manuscript is well constructed, and I believe the study is sound. I recommend publication after minor revisions. Specifically:

Comment 1: Figure 1 appears to be missing from this manuscript.

Response 1: Thank you for pointing that out. There was an issue with the image formatting, which has now been resolved. Figure 1 has been properly adjusted and should now appear in the manuscript.

Comment 2: In the study design in the manuscript, "The efficacy of Nirsevimab against various severity criteria for LRTI related to RSV was evaluated over two seasons: 2022-2023 (without receiving Nirsevimab) and 2023-2024 (receiving Nirsevimab) (Table 3; Figure 3)." While both periods are classified as post-COVID, the circulating variants during each season may be different, which could act as a confounding variable in assessing the severity of RSV and, consequently, the efficacy of the drug. This variation could potentially affect the results of the drug's efficacy. It may be worth commenting on this potential limitation in the manuscript.

Response 2: Thank you for your valuable comment. We agree that the variability in circulating RSV variants between the two post-COVID seasons could act as a confounding factor in assessing the severity of RSV and the efficacy of Nirsevimab. In response to your suggestion, we have revised the manuscript and addressed this point in the discussion section. We now highlight that one of the key strengths of our study is the analysis of four seasons with different RSV infection rates, especially in the context of the COVID-19 pandemic. We emphasize how viral interactions, competition for ecological niches, and non-pharmaceutical interventions impacted viral epidemiology during the pandemic. Additionally, we acknowledge the unpredictability of post-pandemic seasons, which adds value to our findings.

However, we also recognize the potential limitation of varying RSV strains across seasons and how this could influence the results. This aspect has now been included in the discussion to provide a more complete interpretation of the drug’s effectiveness.

Reviewer 2 Report

Comments and Suggestions for Authors

1) The study used a person-time approach, which is appropriate for analyzing time-dependent events such as hospitalization. This method accounts for both the duration and the risk exposure period, making it ideal for studying hospitalizations over different seasons. However, it's worth noting if the study adequately addressed potential time-varying confounders that might affect outcomes over the duration of the study.

2) Suggestion: It would be helpful if the study specified which central tendency measures were used for non-normally distributed data, such as the median. Did the authors also provide effect sizes for any observed differences?

3)Suggestion: It would be insightful to provide more details on how ties or extreme outliers in non-parametric tests were handled, as these can sometimes skew results.

4)It would be helpful to include whether any corrections (such as Bonferroni correction) were applied for multiple comparisons to reduce the risk of Type I errors.

5)The efficacy of Nirsevimab was calculated as ( 1 − point estimate ) × 100 , a standard method in clinical studies to estimate vaccine or treatment efficacy. The inclusion of confidence intervals (CI) adds robustness to the efficacy estimation.  A sensitivity analysis or subgroup analysis could further strengthen the conclusions about efficacy across different population segments, age groups, or underlying conditions.

6)While the efficacy percentages are impressive, it would help to include a more detailed breakdown of the baseline characteristics for the two age groups (under 3 months and 3-6 months) to clarify any potential differences in underlying risk factors. Were the baseline risks of RSV hospitalization similar between these two age groups?

7)It would be helpful to explain how the absence of events was handled statistically. Were any imputation methods used, or were these cases excluded from certain analyses? Additionally, a sensitivity analysis to account for low event rates in such categories could add robustness to the findings.

8)It would be helpful to mention how overdispersion was checked in the Poisson models. If overdispersion was detected, was any alternative model considered, such as negative binomial regression, to account for the potential variability in the data?

Author Response

Reviewer 2

Comment 1: The study used a person-time approach, which is appropriate for analyzing time-dependent events such as hospitalization. This method accounts for both the duration and the risk exposure period, making it ideal for studying hospitalizations over different seasons. However, it's worth noting if the study adequately addressed potential time-varying confounders that might affect outcomes over the duration of the study.

Response 1: Thank you for your insightful feedback. We acknowledge that time-varying confounders could potentially affect the outcomes over the study duration. This limitation has been addressed in the discussion section.

We also investigated the association of time by adding the month and year of admission as a variable to adjust on the model; however, this association was not found to be statistically significant. Consequently, we did not adjust our model with this variable. While additional analyses, such as mediation analysis, could further evaluate confounding factors, time constraints due to the urgency of publishing these critical findings have limited our ability to conduct them. It is important to note that while statistical criteria may be insufficient to fully characterize confounding factors, our analysis showed that adding the time variable did not lead to more than a 10% change in the hazard ratios, indicating that these confounders are not considered substantial. Nonetheless, we will consider conducting more detailed analyses in future studies to enhance our understanding of these dynamics.

Comment 2: It would be helpful if the study specified which central tendency measures were used for non-normally distributed data, such as the median. Did the authors also provide effect sizes for any observed differences?

Response 2: Thank you for your valuable suggestion. In the Materials and Methods section, we specified the treatment of the data and the tests conducted for non-normally distributed data: “A preliminary descriptive analysis of patient characteristics was performed, calculating measures of central tendency and dispersion for quantitative variables, along with counts and percentages for qualitative variables. Differences between groups were assessed using normality tests (Shapiro-Wilk, Kolmogorov-Smirnov) and homogeneity tests (Levene), with parametric (ANOVA, Student's t-test) or non-parametric (Kruskal-Wallis, Wilcoxon rank-sum) tests applied as appropriate. For qualitative variables, χ² or Fisher's exact tests were utilized”.

Additionally, we acknowledge that effect sizes are an important aspect to consider. However, since the descriptive evaluation presented in Table 1 was not central to addressing the initial research question—unlike the results presented in the other tables and figures—we did not elaborate extensively on the differences between groups in Table 1. We appreciate your feedback and will consider including effect sizes in future analyses for a more comprehensive understanding of the observed differences.

Comment 3: It would be insightful to provide more details on how ties or extreme outliers in non-parametric tests were handled, as these can sometimes skew results.

Response 3: We appreciate your attention to the handling of ties or extreme outliers in non-parametric tests. All relevant information regarding the non-parametric tests is explained in the Materials and Methods section of the manuscript.

However, it is important to note that there were few variables in our study that could present extreme outliers, as the majority of our analyses focused on categorical variables. The limited use of continuous variables reduces the likelihood of significant skewing effects from ties or extreme outliers.

Comment 4: It would be helpful to include whether any corrections (such as Bonferroni correction) were applied for multiple comparisons to reduce the risk of Type I errors.

Response 4: We appreciate your recommendation to clarify whether corrections for multiple comparisons were applied. While this information was not initially included in the Materials and Methods section, we conducted for all analysis Bonferroni correction. Now this information appear in Material and Methods: “To reduce the risk of Type I errors due to multiple comparisons, the Bonferroni correction was applied where necessary”.

Comment 5: The efficacy of Nirsevimab was calculated as ( 1 − point estimate ) × 100 , a standard method in clinical studies to estimate vaccine or treatment efficacy. The inclusion of confidence intervals (CI) adds robustness to the efficacy estimation.  A sensitivity analysis or subgroup analysis could further strengthen the conclusions about efficacy across different population segments, age groups, or underlying conditions.

Response 5: Thank you for your insightful comments regarding the efficacy calculations. We have indeed conducted subgroup analyses based on age, which are detailed in Table 3 of the manuscript. We believe that by presenting the efficacy estimates along with confidence intervals and segregating the data by age groups, we have provided a robust and reliable answer to the study's research question. This approach enhances the reliability of our findings and addresses the variations in efficacy across different population segments.

Comment 6: While the efficacy percentages are impressive, it would help to include a more detailed breakdown of the baseline characteristics for the two age groups (under 3 months and 3-6 months) to clarify any potential differences in underlying risk factors. Were the baseline risks of RSV hospitalization similar between these two age groups?

Response 6: Thank you for your suggestion. Upon reviewing the baseline characteristics between the two age groups (under 3 months and 3-6 months), we found no significant differences in the underlying risk factors for RSV hospitalization. Below is a summary of the key comorbidities:

Item

<3 meses (N=281)

3-6 meses (N=188)

p-value

Comorbidities

Neonatal respiratory problems

29 (10.3%)

16 (8.5%)

0.485

APLV

2 (0.7%)

1 (0.5%)

1.000

Atopy

1 (0.4%)

1 (0.5%)

1.000

Cardiopathies

7 (2.5%)

1 (0.5%)

0.214

Syndromes

1 (0.4%)

0 (0%)

1.000

Immunodeficiencies

2 (0.7%)

1 (0.5%)

1.000

Twins

3 (1.1%)

2 (1.1%)

1.000

Premature

11 (3.9%)

12 (6.4%)

0.320

Low birth weight

7 (2.5%)

7 (3.7%)

0.623

As shown, the p-values indicate no significant differences between these two age groups in terms of comorbidities, supporting the conclusion that the baseline risks of RSV hospitalization were similar across both groups.

Comment 7: It would be helpful to explain how the absence of events was handled statistically. Were any imputation methods used, or were these cases excluded from certain analyses? Additionally, a sensitivity analysis to account for low event rates in such categories could add robustness to the findings.

Response 7: We did not use any imputation methods in our study, as there are various approaches, many of which can negatively impact the conclusions, such as the Last Observation Carried Forward (LOCF) method. Instead, data without events were simply excluded from certain analyses by assigning NA (not applicable) values in the relevant tables and figures, including Table 3 and Figure 3. We believe this approach is more suitable for maintaining the integrity of our findings. Additionally, we did not conduct a sensitivity analysis for low event rates, as the number of events was already low across specific categories, making it challenging to derive meaningful insights. The primary focus of our analysis was to assess the overall efficacy of Nirsevimab, and we believe the current results adequately addresses our research questions.

Comment 8: It would be helpful to mention how overdispersion was checked in the Poisson models. If overdispersion was detected, was any alternative model considered, such as negative binomial regression, to account for the potential variability in the data?

Response 8: We initially considered using a quasi-Poisson model to account for potential overdispersion in our data. However, upon evaluation, we found no evidence of overdispersion in any of the cases analyzed. Consequently, we did not pursue this alternative model. We appreciate your input and will ensure to clarify this process in the manuscript.

Reviewer 3 Report

Comments and Suggestions for Authors

In the current manuscript, the authors present data obtained from a retrospective study of RSV hospitalizations in children under six months of age, between October 1st and March 31st, across four seasons in Spain: pre-COVID (2018-2019), COVID (2019-2020), post-COVID, pre-Nirsevimab availability (2022-2023), and post-Nirsevimab availability (2023-2024). The data show that, following the introduction of Nirsevimab, there was a substantial reduction in RSV-related lower respiratory tract infection hospitalizations in infants under six months of age.

The manuscript focuses on a clinically relevant topic that will be of high interest to readers. RSV is responsible for a large burden of severe respiratory illness in young children worldwide. Therefore, interest regarding clinical approaches to prevent RSV infection in children is high. And while several clinical trials have demonstrated the efficacy of Nirsevimab against RSV disease in young children, this manuscript provides needed evidence regarding Nirsevimab performance in real-world clinical settings.

The methodology is described clearly and with sufficient detail. Similarly, the results of the study are presented clearly, with tables and charts used effectively to illustrate trends in RSV-related LRTI hospitalizations in children over the four seasons analyzed. The conclusion summarizes the major findings and highlights the importance of Nirsevimab introduction in combating RSV morbidity and mortality in young children worldwide.

Specific comments:

1. The authors remark that the average Nirsevimab coverage rate in newborns in Spain during the 2023-204 RSV season was 91.9% (line 242-243). Accordingly, do the authors have any data regarding Nirsevimab administration in the 77 patients analyzed during the Nirsevimab season (2023-2024).

2. For each season analyzed a proportion of patient population is made up of premature infants (Table I). There was a slightly highly frequency of premature infants during the 2023-2024 season (7.8%). It would be important to know whether these individuals received palivizumab since its use could also contribute to an observable reduction in hospitalization rates.

3. Similarly, the authors should comment on the potential impact (or lack thereof) of the availability of the maternal RSV vaccine on child hospitalization and severity during the 2023-2024 season. Do data exist on the percentage of pregnant people in Spain that received RSV vaccine during the 2023–2024 RSV season?

4. Line 198: since it appears that the authors do not have information on which study patients received Nirsevimab, the wording used in this section needs to be modified. Without information on which infants during the 2023-2024 season were immunized, conclusions cannot be made regarding the efficacy of the Nirsevimab. Rather, the data provide insights on the impact of Nirsevimab introduction on the severity RSV hospitalizations in young children. I suggest the modified wording of “efficacy of Nirsevimab introduction” throughout this section.

5. Table 3: the spelling of effectiveness needs to be corrected in the header row.

6. Line 247: the study is not designed nor powered to assess Nirsevimab efficacy. The wording needs to be adjusted accordingly (i.e., “This study aims to provide a more accurate picture of the impact of Nirsevimab introduction on RSV-related hospitalizations and severity of RSV-related hospitalizations…”.

7. Line 280: another limitation is that the Nirsevimab status of infants during the 2023-2024 season is unknown.

Author Response

Reviewer 3

Comment 1: The authors remark that the average Nirsevimab coverage rate in newborns in Spain during the 2023-2024 RSV season was 91.9% (line 242-243). Accordingly, do the authors have any data regarding Nirsevimab administration in the 77 patients analyzed during the Nirsevimab season (2023-2024).

Response 1: Yes, we do have data regarding Nirsevimab administration in the 77 patients analyzed during the 2023-2024 RSV season. We have created a figure that illustrates who received Nirsevimab, along with the reasons for its administration, whether it was administered before or after admission. This figure will be included in the annexes of the manuscript and referenced in the first paragraph of results.

Comment 2: For each season analyzed a proportion of patient population is made up of premature infants (Table I). There was a slightly highly frequency of premature infants during the 2023-2024 season (7.8%). It would be important to know whether these individuals received palivizumab since its use could also contribute to an observable reduction in hospitalization rates.

Response 2: Thank you for your comment regarding the proportion of premature infants in our study. We would like to clarify that none of the premature infants received palivizumab instead of nirsevimab during the 2023-2024 season. This aligns with the Recommendations of the Spanish Society of Neonatology for the prophylaxis against severe respiratory syncytial virus infections with nirsevimab for the 2023-2024 season, which can be found at the following link: https://www.analesdepediatria.org/es-recomendaciones-sociedad-espanola-neonatologia-profilaxis-articulo-S1695403323001844

Additionally, we have included this information in the discussion section, stating that, "… Notably, none of the premature infants received palivizumab during the 2023-2024 season, in accordance with the recommendations for the prophylaxis of severe RSV infections.

Comment 3: Similarly, the authors should comment on the potential impact (or lack thereof) of the availability of the maternal RSV vaccine on child hospitalization and severity during the 2023-2024 season. Do data exist on the percentage of pregnant people in Spain that received RSV vaccine during the 2023–2024 RSV season?

Response 3: Thank you for your insightful question. We would like to clarify that no pregnant individuals received the RSV vaccine during the 2023-2024 season. While vaccination against RSV in pregnant women is approved in Spain, the current public health strategy implemented by the Spanish Ministry of Health is the administration of Nirsevimab to newborns.

Comment 4: Line 198: since it appears that the authors do not have information on which study patients received Nirsevimab, the wording used in this section needs to be modified. Without information on which infants during the 2023-2024 season were immunized, conclusions cannot be made regarding the efficacy of the Nirsevimab. Rather, the data provide insights on the impact of Nirsevimab introduction on the severity RSV hospitalizations in young children. I suggest the modified wording of “efficacy of Nirsevimab introduction” throughout this section.

Response 4: We appreciate your observation regarding the need for clarity in our wording. We indeed have data on which patients received Nirsevimab during the 2023-2024 RSV season. To provide a comprehensive overview, we have created a figure that details the administration of Nirsevimab, including the reasons for its use and whether it was given before or after admission. This figure will be included in the annexes of the manuscript. We will also consider modifying the wording to emphasize the "efficacy of Nirsevimab introduction" throughout this section, as you suggested.

Comment 5: Table 3: the spelling of effectiveness needs to be corrected in the header row.

Response 5: Thank you. Corrected.

Comment 6: Line 247: the study is not designed nor powered to assess Nirsevimab efficacy. The wording needs to be adjusted accordingly (i.e., “This study aims to provide a more accurate picture of the impact of Nirsevimab introduction on RSV-related hospitalizations and severity of RSV-related hospitalizations…”.

Response 6: Thank you for your comment, and we have made the necessary corrections to the wording. Line 247 now reflects that "this study aims to provide a more accurate picture of the impact of the introduction of Nirsevimab on RSV-related hospitalizations and the severity of RSV-related hospitalizations."

Comment 7: Line 280: another limitation is that the Nirsevimab status of infants during the 2023-2024 season is unknown.

Response 7: Thank you for your comment regarding Line 280. We want to clarify that we do indeed have data on which patients received Nirsevimab during the 2023-2024 RSV season. To provide a comprehensive overview, we have created a figure that details the administration of Nirsevimab, including the reasons for its use and whether it was given before or after admission. This figure will be included in the annexes of the manuscript and referenced in the first paragraph of results.